# Interface Optimization and Transport Modulation of Sm_2_O_3_/InP Metal Oxide Semiconductor Capacitors with Atomic Layer Deposition-Derived Laminated Interlayer

**DOI:** 10.3390/nano11123443

**Published:** 2021-12-19

**Authors:** Jinyu Lu, Gang He, Jin Yan, Zhenxiang Dai, Ganhong Zheng, Shanshan Jiang, Lesheng Qiao, Qian Gao, Zebo Fang

**Affiliations:** 1School of Materials Science and Engineering, Anhui University, Hefei 230601, China; lu19971210@outlook.com (J.L.); B01914070@ahu.edu.cn (J.Y.); B19201052@ahu.edu.cn (L.Q.); gaoq@ahu.edu.cn (Q.G.); 2School of Physics and Optoelectronics Engineering, Anhui University, Hefei 230601, China; physdai@ahu.edu.cn; 3School of Integration Circuits, Anhui University, Hefei 230601, China; jiangshanshan@ahu.edu.cn; 4School of Mathematical Information, Shaoxing University, Shaoxing 312000, China

**Keywords:** MOS capacitors, Sm_2_O_3_ high-k gate dielectric, atomic layer deposition, conduction mechanisms, interface state density

## Abstract

In this paper, the effect of atomic layer deposition-derived laminated interlayer on the interface chemistry and transport characteristics of sputtering-deposited Sm_2_O_3_/InP gate stacks have been investigated systematically. Based on X-ray photoelectron spectroscopy (XPS) measurements, it can be noted that ALD-derived Al_2_O_3_ interface passivation layer significantly prevents the appearance of substrate diffusion oxides and substantially optimizes gate dielectric performance. The leakage current experimental results confirm that the Sm_2_O_3_/Al_2_O_3_/InP stacked gate dielectric structure exhibits a lower leakage current density than the other samples, reaching a value of 2.87 × 10^−6^ A/cm^2^. In addition, conductivity analysis shows that high-quality metal oxide semiconductor capacitors based on Sm_2_O_3_/Al_2_O_3_/InP gate stacks have the lowest interfacial density of states (*D*_it_) value of 1.05 × 10^13^ cm^−2^ eV^−1^. The conduction mechanisms of the InP-based MOS capacitors at low temperatures are not yet known, and to further explore the electron transport in InP-based MOS capacitors with different stacked gate dielectric structures, we placed samples for leakage current measurements at low varying temperatures (77–227 K). Based on the measurement results, Sm_2_O_3_/Al_2_O_3_/InP stacked gate dielectric is a promising candidate for InP-based metal oxide semiconductor field-effect-transistor devices (MOSFET) in the future.

## 1. Introduction

As the integration of IC continues to increase, CMOS feature sizes will also continue to decrease in order to reduce the cost of individual transistors, increase the switching speed of transistors, and reduce the power consumption of the circuit. As the feature size of CMOS devices continues to decrease, it will also cause the SiO_2_ gate dielectric and Si substrate used in conventional processes to decrease in size as well. When the size is smaller than a certain limit, the gate leakage current will grow exponentially, while the device will not operate properly due to the laws of quantum physics [1,2]. The use of high-k materials for the replacement of SiO_2_ gate dielectrics is an option that has been shown to be feasible [3]. Among these high k materials, samarium oxide (Sm_2_O_3_) is considered as the next potential gate dielectric due to its high dielectric constant (~15) [4], sufficiently large band gap (5.1 eV) [5], low hygroscopicity, and high chemical and thermal stability [6]. Coulomb scattering and phonon scattering at the interface between the high-k gate dielectric and the channel material lead to a significant reduction in channel mobility, which severely affects the further increase in the speed of CMOS logic devices. Selecting channel materials with high mobility is an effective way to solve this problem [7]. Compared with conventional Si-based material CMOS devices, III-V group semiconductors have advantages due to their large switching speed and small dynamic power consumption [8]. Among the group III-V semiconductors, InP has received more attention due to its higher carrier mobility and smaller band gap [9].

However, InP is prone to the formation of interfacial defects, which can limit the operating performance of the device [10]. Also, a surface with many chemical impurities can have a considerable impact on the performance of InP MOS capacitors [11]. High *D*_it_ leads to the frequency dispersion of the Fermi energy level pegging and capacitance, which also prevents the formation of inverse or accumulation layers in CMOS devices [12].

Different InP surface passivation methods have been investigated for a long time, including low-temperature processes [13], ozone treatment [14], chemical etching [15], and sulfide solution passivation [16,17]. A great deal of work has also been devoted to atomic layer deposition (ALD) passivation layers to modulate the InP interface [18]. It has been demonstrated that ALD-derived Al_2_O_3_ films can effectively suppress the interfacial diffusion from the substrate to the high-k films. More importantly, the operating temperature can be kept low (~200 °C) when the Al_2_O_3_ film is on the passivated substrate surface. There are previous reports confirming that the insertion of Al_2_O_3_ between the high-k gate dielectric and GaAs can improve the thermal stability. However, even in the presence of an Al_2_O_3_ passivation layer to improve the interface, the diffusion of In and P elements into the gate dielectric still has an impact on the electrical characteristics of the device when fabricating InP MOS capacitors. R. V. Galatage et al. reported that the In-O and P-O states at the interface lead to a degradation of the electrical characteristics [19]. Their results also demonstrated the effectiveness of ALD-derived Al_2_O_3_ passivation layers between the gate dielectric and the InP substrate. Chee-Hong An et al. systematically analyzed that Al_2_O_3_ can inhibit dissociation and reactant diffusion in InP substrates [20]. However, the effect of the position of the Al_2_O_3_ passivation layer on the electrical properties and interfacial bonding state of InP MOS devices has not been reported systematically.

In this work, we deposited Sm_2_O_3_ films by magnetron sputtering and obtained Al_2_O_3_ passivation layers by ALD equipment to fabricate three different gate stacks on InP substrates, corresponding to Al_2_O_3_/Sm_2_O_3_/InP, Al2_2_O_3_/Sm_2_O_3_/Al_2_O_3_/InP, and Sm_2_O_3_/Al_2_O_3_/InP, respectively. X-ray photoelectron spectroscopy (XPS) and electrical measurements were used to investigate the effect of Al_2_O_3_ passivation position on the chemical composition and electrical parameters of the interface. In addition, the leakage current conduction mechanisms (CCMs) of InP-based MOS capacitors with three different laminated gate electrical stacks measured at room temperature and low temperature (77–227 K) were systematically investigated.

## 2. Materials and Methods

In this work, we chose sulfur-doped n-type InP wafers as the substrate for fabricating MOS capacitors. Before depositing the Sm_2_O_3_ gate dielectric, the wafers were subjected to a standard degreasing process by sequential immersion in ethanol and acetone for 5 min each. After that, the wafers were immersed in 20% ammonium sulfide solution for 15 min to remove the native oxides. Then, the wafers are rinsed with deionized water and then blown dry with high purity nitrogen gas. The cleaned wafers are transferred to an ALD system (MNT-PD100Oz-L6S1G2, MNT Micro and Nanotech). On the ALD process, plasma O_2_ and trimethylaluminum (TMA) were selected as the oxidant and aluminum metal precursor, and a 2 nm Al_2_O_3_ passivation layer was deposited on the InP substrate. The Al_2_O_3_ passivation layers were deposited by using 30 pulse cycles of plasma O_2_ precursors [O_2_(2s)/Ar purge (25s)] and 15 pulse cycles of trimethylaluminum (TMA) and plasma O_2_ [TMA (0.03s)/O_2_ 2s/Ar purge (25s)], respectively. During this process, the chamber pressure and the deposition temperature were maintained at 35 Pa and 200 °C. After ALD Al_2_O_3_ passivation, the wafers were transferred to a sputtering chamber to deposit Sm_2_O_3_ gate dielectrics by sputtering samarium target with purity of 99.9%. When the chamber pressure was 0.8 Pa, Sm_2_O_3_ thin film was deposited under an Ar/O_2_ (50/10 sccm) atmosphere. To explore the electrical characteristics of Sm_2_O_3_/InP MOS capacitors with different stacking positions of Al_2_O_3_ passivation layers, a 200-μm-diameter Al electrode was deposited by thermal evaporation, while an aluminum electrode was grown on the back side to form an ohmic contact. Figure 1 demonstrates the schematics of InP-MOS capacitors based on different stacked gate dielectric structures. Sample S1 corresponds to Al_2_O_3_ (2 nm)/Sm_2_O_3_ (8 nm)/InP, sample S2 corresponds to Al_2_O_3_ (2 nm)/Sm_2_O_3_ (6 nm)/Al_2_O_3_ (2 nm)/InP, and sample S3 corresponds to Sm_2_O_3_ (8 nm)/Al_2_O_3_ (2 nm)/InP, respectively. By using the ESCALAB 250Xi system, XPS (X-ray photoelectron spectroscopy) measurements were performed at Al Ka (1486.7 eV) to investigate the interfacial chemical properties of the Sm_2_O_3_/InP gate stack and the chemical function of the Al_2_O_3_ passivation layer. Furthermore, the escape angle used in obtaining the XPS profiles is 50° and the corresponding probing depth is about 1–10 nm. Ultraviolet-visible spectroscopy (Shimadzu, UV-2550) was performed to obtain the samples’ optical band gap. The physical thickness of the above samples was extracted by using spectroscopic ellipsometry measurements (SANCO Inc., Shanghai, China, SC630) with the help of the Cauchy-Urbach model. The Cascade Probe Station was connected to the semiconductor analysis equipment (Agilent B1500A) for capacitance-voltage (C-V), transconductance-voltage (G-V), and leakage current-voltage (I-V) measurements at room temperature. For low temperature (77–227 K) leakage current testing, the Lake Shore Cryotronics Vacuum Probing Station was used. 

## 3. Results and Discussion

### 3.1. XPS Analyses

To evaluate the chemical bonding states of various stacked gate dielectrics, XPS measurements were carried out. Figure 2 displays the In 3d, P 2p, and O 1s XPS spectra of three samples with various stacked gate dielectrics. It can be noted that In 3d spectra can be deconvoluted into four components that represent the InP, InPO_4_, In(PO_3_)_3_, and In_2_O_3_, respectively. The relative intensity values of the different components have been extracted and are shown in Figure 3a. For S2 and S3, the contents of In(PO_3_)_3_ and In_2_O_3_ shows a decreasing trend, indicating that the ALD-derived Al_2_O_3_ passivation layers prior to Sm_2_O_3_ deposition can significantly prohibit the formation of In and P suboxides, which can be attributed to the interface cleaning function of plasma O_2_ [21]. Compared to S2, the peak areas of InPO_4_ corresponding to S1 and S3 remain approximate at about 7.89% and 5.20%, which is much lower than that of S2 (19.41%), indicating that double deposition of ALD-derived Al_2_O_3_ may accelerate the diffusion of oxygen in the substrate and the formation of indium phosphate. During the secondary deposition of Al_2_O_3_, more oxygen vacancies may generate, which can be ascribed to plasma O_2_ acting as an oxygen source, and promote the oxygen interdiffusion between Al_2_O_3_ passivation layers and the InP substrate. In(PO_3_)_3_ can react with In to produce InP and InPO_4_ using the following reaction Equation [22].
8In + 4In(PO_3_)_3_ → 3InP + 9InPO_4_(1)

More importantly, sample S3 shows a tendency to decrease the content of In(PO_3_)_3_ and AlPO_4_ compared to sample S2, which can give a detailed illustration from the phenomenon of P 2p spectral changes. As shown in Figure 2b, it can be noted that P 2p spectra can be deconvoluted into four components, which represent InP, InPO_4_, In(PO_3_)_3_, and AlPO_4_, respectively. No P_2_O_5_ was detected in all samples, which can be attributed to the fact that gaseous P_2_O_5_ generated during the deposition can easily diffuse through the defects in the gate dielectric [22]. The peak area ratio of In(PO_3_)_3_ for S2 and S3 showed a significant decreasing trend compared to S1, indicating that Al_2_O_3_ prior to the deposition of Sm_2_O_3_ gate dielectric can inhibit the formation of P-O bound states and improve the interfacial quality. The detection of AlPO_4_ in P 2p spectra can be attributed to the reaction equation described below [23].
4Al + 7O_2_ + 2In(PO_3_)_3_ + 2InP → 4AlPO_4_ + 4InPO_4_(2)

Based on the mentioned reaction above, it can be inferred that two depositions of Al_2_O_3_ passivation layers increase the formation of AlPO_4_, which is confirmed by the change in peak area ratio shown in Figure 3c. In order to systematically explore the interfacial chemistry of various stacked gate dielectrics, O 1s spectra were investigated and are shown in Figure 2c. O 1s spectra can be deconvoluted into Sm_2_O_3,_ Al_2_O_3,_ InPO_4_, In(PO_3_)_3_, and AlPO_4_. According to the reaction Equation (2), in the plasma O_2_ atmosphere, In(PO_3_)_3_ can react with O_2_ to produce AlPO_4_ and InPO_4_ and leads to the disappearance of In(PO_3_)_3._ In agreement with the previous In 3d and P 2p spectra, S1 has the largest In(PO_3_)_3_ content, leading to a decrease in interfacial quality and deterioration of electrical properties. Meanwhile, AlPO_4_ of S2 is the highest, originating from the second deposition of Al_2_O_3_. For S3 sample, the contents of InPO_4_, AlPO_4_, and In_2_O_3_ were significantly controlled, indicating that the addition of an ALD-derived Al_2_O_3_ layer prior to the deposition of Sm_2_O_3_ gate dielectric could reduce the generation of suboxides and improve the interfacial quality. 

### 3.2. Band Alignment Characteristics

To assess the optical characteristics of the three various stacked gate dielectrics, UV-Vis spectroscopy was used to obtain the absorption spectra and the optical bandgap values (Figure 4) of samples S1, S2, and S3 were determined to be 5.49, 5.51, and 5.63 eV, respectively, based on the Tauc relationship [24]. Compared with pure Sm_2_O_3_ and pure Al_2_O_3_, the band gaps of three various stacked gate dielectrics showed a value balance [25]. Also, this section investigates the valence band maximum (VBM) of various stacked gate dielectrics, as the valence band alignment is crucial for assessing the interface quality. As shown in Figure 5a, the band gap values of InP substrates were derived from XPS measurements, while the valence bands of samples S1, S2, and S3 were deduced from the absorption spectra by linear extrapolation. Based on Kraut’s method [26], we also calculated the valence band shift ($∆E_{V}$) to evaluate the valence band electronic structure of the samples. By using Sm 3d_5/2_ and In 3d core-level spectra, the $∆E_{V}$ of high-k/InP gate stacks was determined based on the following formula:(3)$$∆E_{V}={\left(E_{\mathrm{In}\;3d}-E_{V}\right)}_{\mathrm{InP}}-{\left(E_{\mathrm{Sm}\;3d}-E_{V}\right)}_{\mathrm{high}-k}-{\left(E_{\mathrm{In}\;3d}-E_{\mathrm{Sm}\;3d}\right)}_{\mathrm{high}-k/\mathrm{InP}}$$
where E_In 3d_ (InP) and E_Sm 3d_ (high-k) corresponding to the core-level positions are extracted to be 445.8 and 1084.3 eV. In addition, the E_v_ (InP) and E_v_ (high-k) represent the VBM (Valence-Band Maximum) of the bulk materials. The values of ΔE_v_ are calculated as 1.87, 1.82, and 1.76 eV, respectively, based on the binding energy difference in the high-k/InP structure. Meanwhile, the value of the conduction band offset (ΔE_c_) is obtained by subtracting the extracted ΔE_v_ and the band gap of the InP (1.34 eV) from the band gap of dielectric layers [27].
(4)$${\Delta E}_{C}{\;(\mathrm{high}-k/\mathrm{InP})=E}_{g}\;(\mathrm{high}-k)-{\Delta E}_{V}\left(\mathrm{High}-k/\mathrm{InP}\right)-E_{g}\left(\mathrm{InP}\right)$$

As shown in Figure 5b, the ΔE_c_ values for the three samples were calculated as 2.28, 2.35, and 2.53 eV. According to previous reports in the literature, ΔE_c_ is related with the tunneling leakage current. The higher ΔE_c_ indicates that the leakage current of the Sm_2_O_3/_Al_2_O_3_/InP samples is smaller. 

### 3.3. Electrical Properties of InP-MOS Capacitors

#### 3.3.1. Capacitance-Voltage Measurements

The frequency dependent capacitance-voltage curves of sample S1, S2, and S3 with double sweep mode are shown in Figure 6a–c. When the frequency increases, all samples show a decreased accumulation capacitance. 

Meantime, at high frequency conditions, the series resistance will deviate from the predetermined theoretical value due to the disappearance of the interface trap charge. On the contrary, at low frequencies, when the oxide capacitance (C_ox_) connects with the space charge capacitance (C_sc_), the value of the accumulation region increases with the series resistance due to the interface state showing frequency-dependent properties [28,29,30]. 

The decrease in the accumulation capacitance can be attributed to the fact that the interfacial traps do not have enough time to respond to the voltage frequency [30]. The maximum accumulation capacitance and minimum hysteresis voltage were observed in the S3 sample, indicating that the Al_2_O_3_ passivation layer suppressed the appearance of In and P oxides and the formation of low-K interfacial layers. To evaluate the interface quality, important electrical parameters such as equivalent oxide thickness (EOT), dielectric constant (k), flat band voltage (V*_fb_*), hysteresis voltage (ΔV*_fb_*), oxidation charge density (*Q_ox_*), and boundary trapped oxide charge density (*N_bt_*) were extracted from the test curves, and these data are presented in Table 1. The variation of V*_fb_* depends on the values of oxide capacitance and bulk oxide charge [31]. The k values corresponding to samples S1, S2, and S3 are calculated to be 12.96, 14.39, and 14.75, which are consistent with the previous investigation [4]. 

A small V*_fb_* of 0.19 V was observed for sample S3. This phenomenon can be explained by the following statement: electrons are easily captured by oxygen vacancies to form negatively charged interstitial oxygen atoms [32] and as fewer oxygen vacancies exist at the interface, the smaller the positive flat voltage required to maintain the band unbent [33]. Also, the hysteresis voltage depends on the boundary trap caused by the intermixing of the high K layer and the interfacial layer [34]. The value of the hysteresis voltage reaches a minimum (1.55 mV) for S3, indicating that the boundary trapping charge becomes weaker after the insertion of Al_2_O_3_ between the gate dielectric and the substrate. The *Q_ox_* and *N_bt_* values were calculated from the obtained V*_fb_* and ΔV*_fb_* values by the following equations [35].
(5)$$Q_{ox}=-\frac{C_{\max}\left(v_{fb}-\varphi_{ms}\right)}{qA}$$
(6)$$N_{bt}=-\frac{C_{\max}△V_{fb}}{qA}$$
where $\varphi_{ms}$ is the contact potential difference between Al electrode and InP substrate, q is the electronic charge, and A is the Al electrode areas. According to Table 1, it can be noticed that S3 has the lowest *Q_ox_* and *N_bt_*, which implies the reduction of interfacial trap defects and the optimization of interfacial properties. 

#### 3.3.2. Conductivity-Voltage Measurements

Moreover, to quantify the interface defect distribution for all samples, the interface state density (*D_it_*) has been extracted by the conductivity-voltage measurements with frequencies varying from 100 kHz to 1 MHz. *D_it_* is related to the parallel interfacial trap capacitance (C_it_) and parallel conductivity (G). At the same time, C_it_ can be related by the following equation. C_it_ = q*D_it_*, while the condition is that the position of the energy level does not change *D_it_*. The basic principle of conductivity measurements is to analyze the losses due to the diversity of charge states at the trap level. Near the Fermi level, the synchronous conductivity occupancy is mobilized by the interfacial traps to produce a regular variation. The maximum loss occurs when the interface trap is resonantly shifted with the applied AC signal (ωτ = 1). The response time of the characteristic trap changes the frequency, τ = 2π/ω. The capture and emission rates from Shockley-Redhall theory modulate the response time [36]:(7)$$\tau =\frac{exp\left[∆E/k_{B}T\right]}{\sigma v_{th}D_{dos}}$$
where there is an energy difference ∆E between the trap level E_T_ and the edge of the majority carrier band, $v_{th}$ is the majority carrier being thermally activated to obtain the average velocity, *D_dos_* is the effective density of states of the majority carrier band, $k_{B}$ is the Boltzmann constant, and *T* is the temperature [37]. The curves between conductivity (G/ω) and gate voltage for all samples are shown in Figure 7a–c. The apparent shift of the conductivity peak proves the validity of the Fermi-level shift and confirms the existence of the Fermi-level deconvolution effect [38]. Assuming that the underlying surface oscillations can be neglected, the value of *D_it_* is inferred using the normalized parallel conductivity peak ${\left(G_{P}/\omega \right)}_{max}$ [39].
(8)$$D_{it}\approx \frac{2.5}{Aq}{\left(\frac{G_{p}}{\omega }\right)}_{max}$$
where *A* is the device area. It is necessary to confirm the transformation law between the band bending potential of the energy location E_T_ and the trap energy level distribution. Furthermore, the values of E_T_ can be determined by the frequency of ${\left(G_{P}/\omega \right)}_{max}$, where Equation (8) is used to calculate *D_it_* and to correspond its value to ∆E [40].
(9)$$∆E=\left(E_{C}-E_{T}\right)=\frac{k_{B}T}{q}In\left(\frac{\sigma vD_{dos}}{2\pi f_{max}}\right)$$

Figure 7d shows the variation of *D_it_* for the three samples. With the increase of ΔE, the value of *D_it_* shows an increasing trend. However, S3 possesses a lower density of interfacial states compared to S1 and S2, which indicates that the insertion of an Al_2_O_3_ passivation layer between the Sm_2_O_3_ gate dielectric and the InP substrate can suppress the formation of In and P suboxides and improve the quality of MOS capacitors. 

We compared some of the data obtained from this work with some previously published work. As can be seen in Table 2, the Sm_2_O_3_ dielectric has a smaller leakage current density than TiO_2_ and HfO_2_, indicating that the Sm_2_O_3_ stacked gate dielectric has a larger conduction band shift, resulting in an increased barrier height and thus a reduced leakage current density. The Sm_2_O_3_ stacked gate dielectric has the smallest hysteresis value, indicating that the trapped charge in the gate dielectric is not very sensitive to the frequency response of the voltage, and will trap fewer electrons to keep the energy band from being bent, while the device maintains a consistent response to different test voltages in the antipattern region. In the interface state density, it is smaller than HfO_2_ as the gate dielectric directly deposited in InP, but it seems to be higher than TiO_2_ gate dielectric, considering the different testing methods, the interface state density of the current work is obtained directly by conductivity method with accuracy, the previous work is by C–V curve, there may be some differences.

#### 3.3.3. *J*−*V* Analyses and Conduction Mechanisms at Room Temperature

Figure 8a shows the leakage current characteristics of all samples measured at room temperature. The leakage current density (*J*) values for S1, S2, and S3 at 1 V are 1.07 × 10^−5^, 8.42 × 10^−6^, and 2.87 × 10^−6^ A/cm^2^, respectively. It can be seen that S1 has a higher leakage current density, which can be attributed to larger interface traps and the border traps that deteriorate the interface quality and degrade the device performance [44]. For the S3 sample, the minimum leakage current density has been observed, which is due to the higher Δ*E*_c_ and the suppressed tunneling in the Sm_2_O_3_/Al_2_O_3_/InP gate stack [45].

To investigate the leakage current characteristics of various stacked gate dielectrics, we systematically studied three different current conduction mechanisms (CCMs) under substrate injection, as shown in Figure 8b–d. The extracted important electrical parameters are listed in Table 3.

Schottky emission (*SE*) is a typical type of thermal ionization emission in which charges gain energy to overcome barriers to migration into the dielectric. The standard *SE* can be described as [46]:(10)$$J_{SE}=A^{*}T^{2}exp\left[\frac{-q\left(\varphi_{B}-\sqrt{qE/4\pi \varepsilon_{0}\varepsilon_{r}}\right)}{k_{B}T}\right]$$
(11)$$A^{*}=\frac{4\pi qk_{B}^{2}m_{ox}^{}*}{h^{3}}=120\frac{m_{ox}^{}*}{m_{0}}$$
where *A*^*^ is the effective Richardson constant, the free electron mass and the effective mass of electrons in the gate dielectric correspond to m_o_ and m_ox*_, E is the electric field, $q\varphi_{B}$ is the Schottky barrier height, and ε_o_ and ε_r_ represent the vacuum dielectric constant and the optical dielectric constant, respectively [47]. It is observed in Figure 8b that at lower electric fields (0.36–0.81 MV/cm), there is a good linear relationship between ln(J/T^2^) and E^1/2^ for S1, S2, and S3. The slope of the SE diagram is denoted as $\sqrt{q^{3}/{4\pi \varepsilon }_{0}\varepsilon_{r}}/k_{B}T$. The fitted ε_r_ and the refractive index *n* (*n* = ε_r_^1/^^2^) for S1, S2, and S3 are (4, 2), (4.96, 2.23), and (4.23, 2.06), respectively. All the fits are consistent with the previously reported values [48], revealing that CCM (current conduction mechanism) at room temperature is dominated by SE emission in the low electric field region. 

The Poole–Frenkel (*PF*) emission can be ascribed to the thermally excited electrons obtaining sufficient energy to escape from traps into the conduction band of the dielectric at a higher electric field, which can be expressed by the following formula [49]: (12)$$J_{PF}=AE\;exp\left[\frac{-q\left(\varphi_{t}-\sqrt{qE/\pi \varepsilon_{0}\varepsilon_{ox}}\right)}{k_{B}T}\right]$$
where *A* represents a constant, the trap energy level of the conduction band corresponds to $\varphi_{t}$, and $\varepsilon_{ox}$ represents the dielectric constant. According to the previous theory, ln(*J*/*E*) should have a good proportionality with *E*^1/2^, as shown in Figure 8c. The ε_ox_ extracted from the slope of the fitted line for all samples was calculated as 11.90, 13.01, and 13.41, which is in agreement with the reported reference [4]. It can be concluded that at higher electric fields (1.21–1.69 MV/cm), the PF emission dominates the CCM of all samples. Also, the value of the trap energy level ($\varphi_{t})$ can be extracted based on the intercept point of the fitted curve described as $lnB-\frac{q\varphi_{t}}{k_{B}T}$. As shown in Figure 8c, the calculated values of $\varphi_{t}$ are 0.53, 0.54, and 0.55 eV, corresponding to S1, S2, and S3. S3 has the largest $\varphi_{t}$ value in the three samples, indicating that the electrons obtain more energy to cross the trap, leading to present the smallest leakage current density in the S3 sample. 

The high-field dependent conduction mechanism is represented by Fowler-Nordheim tunneling, which is manifested by the fact that the insulating layer can be penetrated by electrons, which enter the conduction band of the gate dielectric in a high electric field. The leakage current density is linked to other parameters of Fowler-Nordheim (*FN*) tunneling and is described by the following Equation [46]:(13)$$J_{FN}=\frac{q^{3}E^{2}}{16\pi^{2}\hbar \varphi_{ox}}\;exp\left[-\frac{4\sqrt{2m_{T}^{*}\varphi_{B}^{3/2}}}{3\hbar qE}\right]$$
where $\varphi_{ox}$ is oxide barrier height; $m_{T}^{*}$ is the tunneling effective electron mass in the gate oxide film, and the other notations remain unchanged from the previous definitions. Figure 8d shows the curve of ln(*J*/*E*2) versus 1/*E*. The slope of the linear fit for the above samples shows an increase in current with increasing electric field, indicating that at high electric fields (1.47–1.85 MV/cm), all three samples are consistent with the *FN* tunneling conduction mechanism. Based on the previous analysis, it can be concluded that all samples are dominated by three main conduction mechanisms. In the lower electric fields, SE emission dominates, however, in the higher electric fields, PF emission dominates together with FN tunneling. 

#### 3.3.4. Low Temperature *J*–*V* Analyses and Conduction Mechanisms

To investigate the variation of CCMs in Sm_2_O_3/_Al_2_O_3_/InP MOS capacitor, low temperature (77–227 K) measurements were performed. The leakage current densities of Sm_2_O_3/_Al_2_O_3_/InP MOS capacitors measured at 1 V were extracted as 4.64 × 10^−9^, 1.48 × 10^−8^, 1.13 × 10^−7^, and 1.02 × 10^−6^ A/cm^2^, corresponding to the temperature range of 77–227 K, respectively. By observing the leakage current densities at different temperatures, the Sm_2_O_3/_Al_2_O_3_/InP gate stack exhibits nearly three orders of magnitude lower leakage current density at 77 K than that measured at room temperature, indicating that the low temperature is favorable for the MOS capacitor to exhibit optimized *J*–*V* characteristics. Figure 9b–d show the variation of the CCM under substrate injection along with the temperature trend. The extracted important electrical parameters are listed in Table 4. Figure 9b shows the fitted lines for the vertical temperature range suitable for SE emission at lower electric fields (0.49–0.90 MV/cm). The extracted important electrical parameters are listed in Table 4. With increasing temperature, the values of $\varepsilon_{r}$ and n calculated from the slope and intercept are (20.39, 4.52), (18.53, 4.31), (10.40, 3.22), and (6.10, 2.47). It can be noted that at extremely low temperature of 77–177 K, these values are completely different from the theoretical values, indicating that SE emission is not the dominant conduction mechanism at lower temperatures. Figure 9c shows the curves in the temperature range 77–227 K compatible with PF emission at higher electric fields (0.64–1.44 MV/cm). Again, it can be noted that $\varphi_{t}$ and ε_ox_ are not in the expected range of values, indicating that the PF emission is not compatible for all samples at intermediate electric fields of 0.64–1.44 MV/cm. FN tunneling is a potential conduction mechanism because of its dependence on the electric field at low temperatures. Figure 9d shows the fitted line of FN tunneling with a temperature range of 77–227 K at higher electric fields (1.11–1.67 MV/cm), and the established slope indicates that FN tunneling is dominant at low temperatures. In conclusion, the effects of SE emission and PF emission are attenuated due to low temperature, and FN tunneling is used to explain the Sm_2_O_3/_Al_2_O_3_/InP stacked gate dielectric structure showing low drain current density. 

Additionally, the integrated dielectric properties in MOS capacitors can be estimated from two important values, including the electron effective mass $m_{ox}^{*}$ and the barrier height $q\varphi_{B}$ [50]. The intercept of the SE emission fitting curve described as $\ln\left(120\frac{m_{0x}^{*}}{m_{0}}\right)-\frac{q_{\varphi_{B}}}{k_{B}T}$ and the slope of the FN tunneling fitting curve expressed as $-6.83\times 10^{7}\sqrt{\left(\frac{m_{T}^{*}}{m_{0}}\right)\varphi_{B}^{3}}$ can be calculated together with the above two values. By setting the equation $m_{0x}^{*}$ = $m_{T}^{*}$, the two key physical quantities $m_{0x}^{*}$ and $q\varphi_{B}$ of Sm_2_O_3_/Al_2_O_3_/InP MOS capacitor are obtained by applying mathematical analysis, which are calculated as 0.23 m_o_ and 0.95 eV, respectively. Figure 10 shows the determination of the electron effective mass and barrier height for S3 sample. The smaller $m_{0x}^{*}$ and the higher $q\varphi_{B}$ are beneficial to obtain better electrical properties and optimized interface quality.

## 4. Conclusions

In this work, we explore in detail the effect of ALD-derived laminated interlayers on the interfacial chemistry and transport properties of sputter-deposited Sm_2_O_3_/InP gate stacks. It has been found that Sm_2_O_3_/Al_2_O_3_/InP gate stack can obviously prevent the diffusion of the substrate diffusion oxide and substantially optimize the electrical properties of MOS capacitors, including a larger dielectric constant of 14.75, a larger accumulation capacitance, and a lower leakage current density of 2.87 × 10^−6^ A/cm^2^. Three different stacked gate dielectric structures are also evaluated by means of conductivity of the interfacial density of states. The results show that the Sm_2_O_3_/Al_2_O_3_/InP stacked gate dielectric achieves the lowest interfacial density of states of 1.05 × 10^13^ cm^−2^eV^−1^. According to the analysis of CCMs, SE emission is dominant in lower electric fields and higher temperature environments, and PF emission as well as F-N tunneling is dominant in higher electric fields. Meanwhile, FN tunneling is the only dominant mechanism at lower temperatures. Also, to evaluate the properties of the whole MOS capacitor in low temperature environment, $m_{0x}^{*}$ and $q\varphi_{B}$ have been determined by a self-consistent method. These findings are of crucial importance for the future fabrication of high mobility InP-based MOSFET (Metal Oxide Semiconductor Field Effect Transistor) devices. 

## Figures and Tables

**Figure 1 nanomaterials-11-03443-f001:**
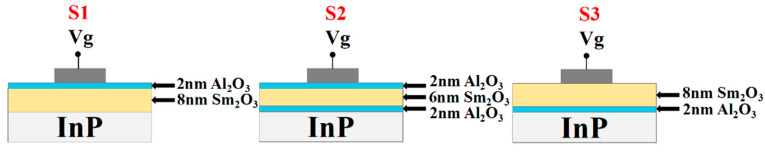
Schematics of InP-based MOS capacitors based on different stacked gate dielectrics.

**Figure 2 nanomaterials-11-03443-f002:**
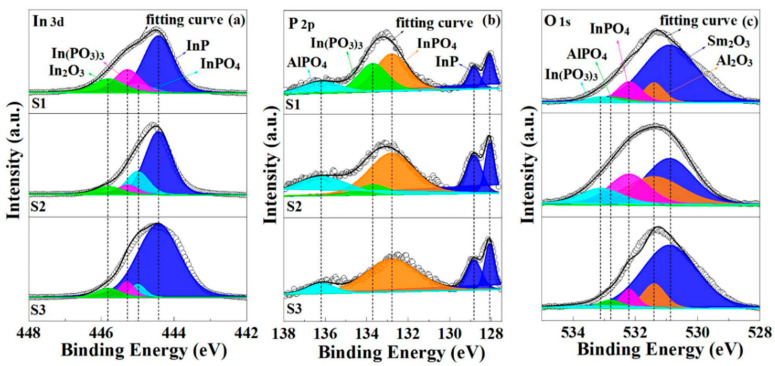
(**a**) In 3d, (**b**) P 2p, and (**c**) O 1s XPS spectra for S1, S2, and S3 sample.

**Figure 3 nanomaterials-11-03443-f003:**
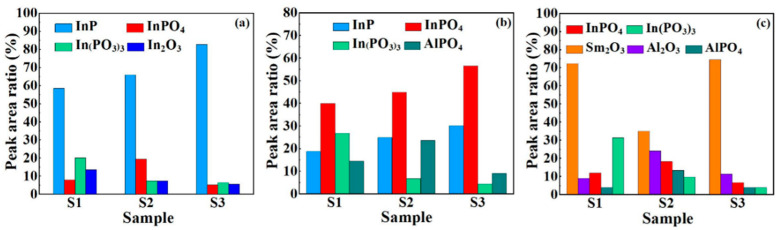
Peak area ratio histograms of (**a**) In 3d, (**b**) P 2p, and (**c**) O 1s spectra.

**Figure 4 nanomaterials-11-03443-f004:**
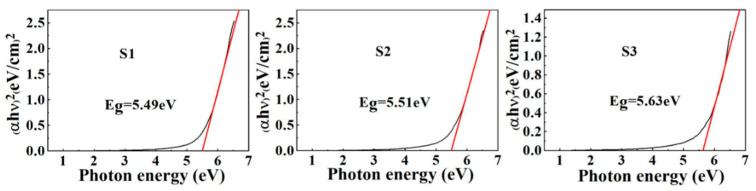
The determination of band gaps for sample S1, S2, and S3.

**Figure 5 nanomaterials-11-03443-f005:**
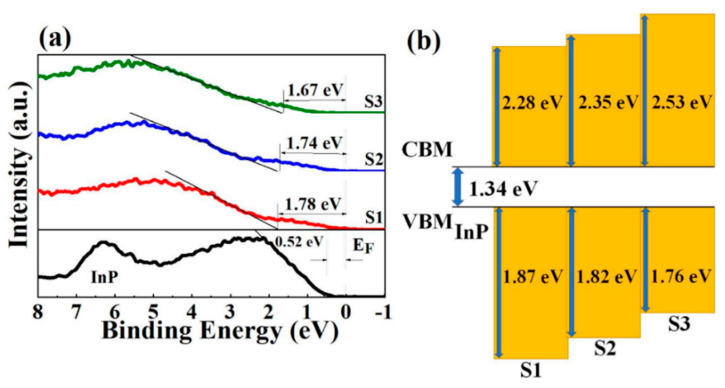
(**a**) Valence band spectra; (**b**) Schematic band diagram of S1, S2, and S3 sample.

**Figure 6 nanomaterials-11-03443-f006:**
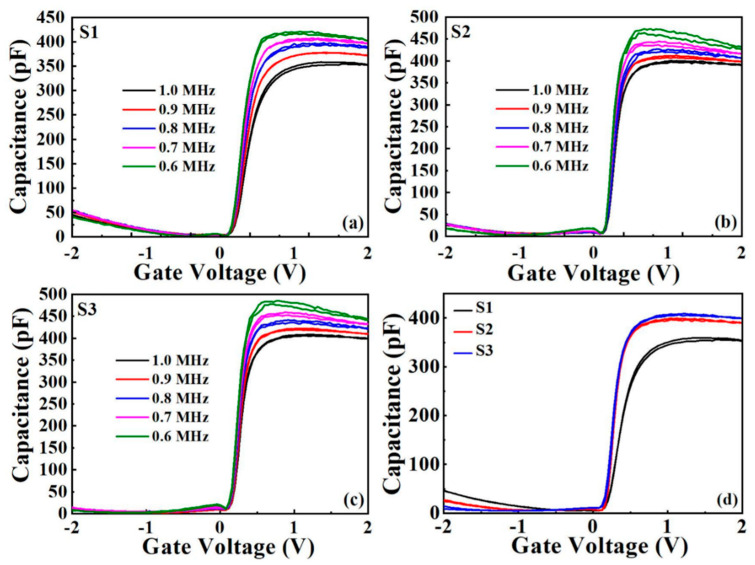
(**a**–**c**) Capacitance–voltage (C–V) curves for S1–S3 measured at different frequency (0.6–1 MHz). (**d**) Capacitance–voltage (C–V) curves for all samples measured at 1 MHz.

**Figure 7 nanomaterials-11-03443-f007:**
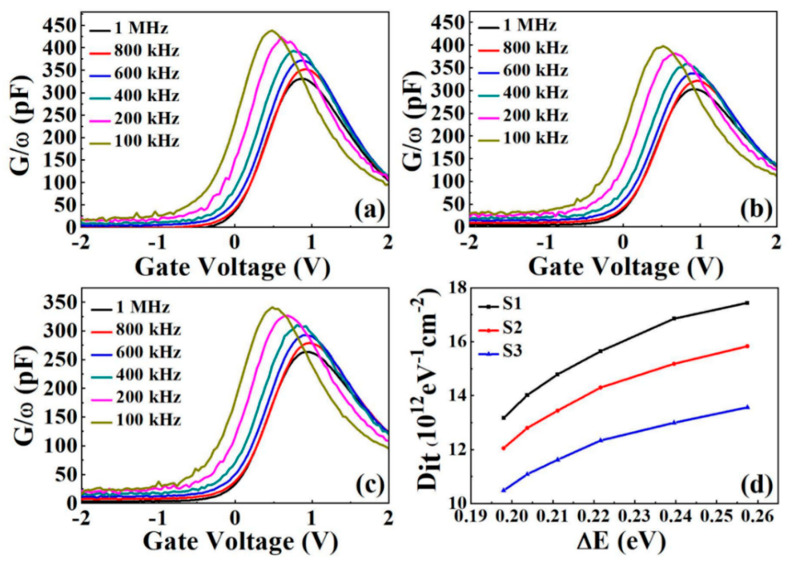
Multi-frequency G–V characteristics of InP-based MOS capacitors of (**a**) S1, (**b**) S2, and (**c**) S3. (**d**) Energy distributions of Dit for S1, S2, and S3.

**Figure 8 nanomaterials-11-03443-f008:**
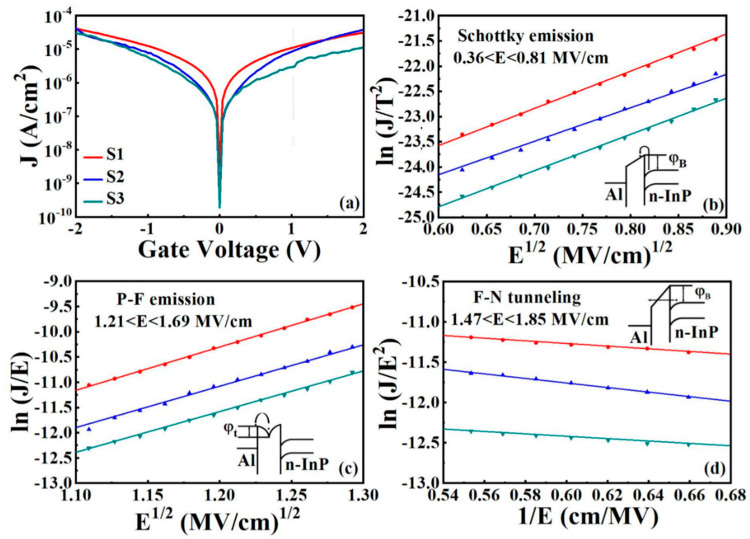
(**a**) J–V characteristics measured at room temperature. (**b**) SE emission, (**c**) PF emission, and (**d**) FN tunneling plots for all the samples under substrate injection.

**Figure 9 nanomaterials-11-03443-f009:**
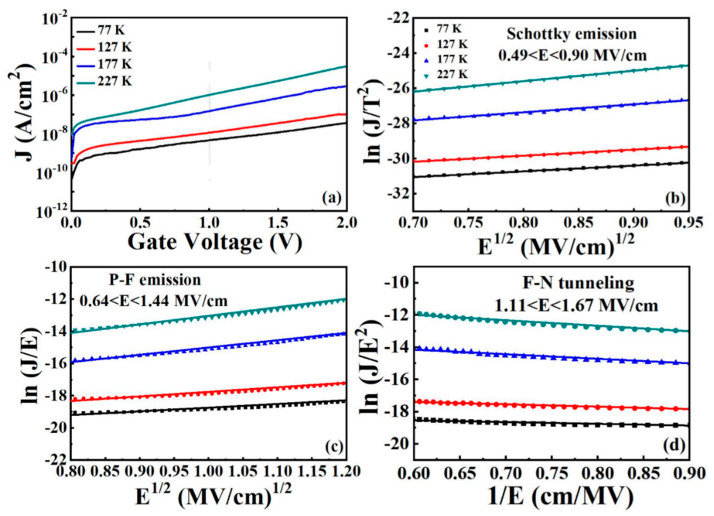
(**a**) J–V characteristics measured at low temperature. (**b**) SE emission, (**c**) PF emission, and (**d**) FN tunneling plots for all the samples under substrate injection.

**Figure 10 nanomaterials-11-03443-f010:**
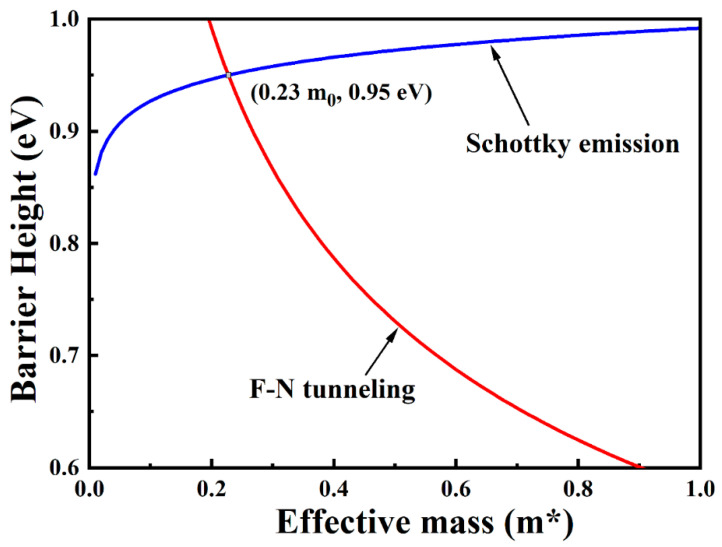
The determination of the electron effective mass and barrier height for S3 sample under substrate injection.

**Table 1 nanomaterials-11-03443-t001:** MOS capacitors electrical parameters obtained from C−V and *J*–*V* Curves.

Sample	EOT (nm)	k	V*_fb_* (V)	△V*_fb_* (mV)	Q_ox_ (cm^−2^)	N_bt_ (cm^−2^)	J (A/cm^−2^)
S1	3.01	12.96	0.25	3.44	−1.62 × 10^12^	−2.46 × 10^10^	1.07 × 10^−5^
S2	2.71	14.39	0.21	5.16	−1.43 × 10^12^	−4.11 × 10^10^	8.42 × 10^−6^
S3	2.65	14.75	0.19	1.55	−1.30 × 10^12^	−1.26 × 10^10^	2.87 × 10^−6^

**Table 2 nanomaterials-11-03443-t002:** Comparison of different InP MOS capacitor parameters.

	Sm_2_O_3_/Al_2_O_3_/InP (This Work)	PMA-TiO_2_/S-InP [41]	TiO_2_/S-InP [41]	10 Å Si IPL/51 Å HfO_2_/InP [42]	70 Å HfO_2_/InP [42]	HfO_2_ (10 nm)/Al_2_O_3_ (0.2 nm)/InGaAs/InP [43]
Leakage current density (A/cm^2^)	2.87 × 10^−6^ at 1 V	1.9 × 10^−7^ at 2 V 2.7 × 10^−5^ at −2 V	5.01 × 10^−6^ at 2 V 1.5 × 10^−2^ at −2 V	1.32 × 10^−3^ at 1 V	3.94 × 10^−2^ at 1 V	2.4 × 10^−2^
k	14.75	39	34	/	/	/
∆Vfb (mV)	1.55	40	250	240	280	/
D_it_ (cm^−2^eV^−1^)	(G-V) 1.05 × 10^13^	(C-V) 3.1 × 10^11^	(C-V) 5 × 10^11^	(C-V) 3-8 × 10^12^	(C-V) 2-9 × 10^13^	(C-V) 2 × 10^12^

**Table 3 nanomaterials-11-03443-t003:** Extracted MOS capacitors electrical parameters measured at room temperature.

Sample	J (A/cm^2^)	ε_r_	*n*	ε_ox_	φ_t_ (eV)
S1	1.07 × 10^−5^	4.00	2.00	11.90	0.53
S2	8.42 × 10^−6^	4.96	2.23	13.01	0.54
S3	2.87 × 10^−6^	4.23	2.06	13.41	0.55

**Table 4 nanomaterials-11-03443-t004:** S3’s MOS capacitors electrical parameters measured at low temperature.

T	J (A/cm^2^)	ε_r_	*n*	ε_ox_	φ_t_ (eV)
77 K	4.64 × 10^−9^	20.39	4.52	164.88	0.54
127 K	1.13 × 10^−8^	18.53	4.31	108.14	0.53
177 K	1.48 × 10^−7^	10.40	3.22	42.66	0.50
227 K	1.02 × 10^−6^	6.10	2.47	31.05	0.47

## Data Availability

The study did not report any data.

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
