# Peer review of "Interface Optimization and Transport Modulation of Sm2O3/InP Metal Oxide Semiconductor Capacitors with Atomic Layer Deposition-Derived Laminated Interlayer"

_nanomaterials, 2021, doi:10.3390/nano11123443_

Round 1

Reviewer 1 Report

Review of the manuscript entitled “Interface Optimization and Transport Modulation of Sm2O3/InP MOS Capacitors with ALD-Derived Laminated Interlayer” for MDPI Nanomaterials.

In the submitted manuscript, the authors present the examination of dielectric stacks composed of sputtered Sm2O3 and ALD Al2O3 on InP. The stacks are characterized by structural, optical and electrical methods. The optimal stack in terms of electrical properties (density of interface states, dielectric constant and leakage current) is identified out of the 3 different versions examined within this work.

While the manuscript contains a thorough investigation of the gate dielectric stacks and its conclusions are based on the results of the analysis performed, I believe that the manuscript is not ready for publication for the following reasons:

  • The English in the manuscript is not very good, something that makes it hard for the readership to follow. Rigorous editing is needed.
  • The authors fail, in my opinion, to provide adequate motivation and explanation of the originality of their work. They mention that their presented results are interesting in terms of making MOSFETs on InP with limited interface traps, a subject of great interest. Unfortunately, they do not show how their results achieve this. For example, the use of other dielectric materials, such as HfO2, result in similar, if not better, results. (for example, reference 12 of the manuscript presents better Dits).

Also, the use of a Al2O3 interface layer as a way to reduce interface states has been used before with similar success (for example, reference 13 of the manuscript).

Not even the use of the dielectric SmO3 is original. References 15 and 17 discuss this material, even though on different substrates.

I believe that the authors should attempt to explain where the originality of their manuscript comes from and how their results advance the subject of better dielectric stacks on InP, by comparing the different electrical results with those of the literature. Perhaps a direct comparison with previously published results will make the advancements proposed in this manuscript easier to see. It might also make it clearer why the SmO3 dielectric is more appropriate for this application than other high-k dielectrics.

Minor remarks:

  • Maybe a more detailed explanation about the deposition of the SmO3 dielectric would be useful. The authors mention that the sputtering target used is a pure Sm target but do not mention the conditions for the sputtering process such as the atmosphere. In order to get the oxide, there must be oxygen in the chamber, at least.

Reviewer 2 Report

The manuscript by J. Lu et al. presents the study of physical properties and electrical characteristics of the metal-oxide semiconductor capacitors with Al2O3 interlayers prepared by atomic-layer deposition technique. The impact of this work is not well pronounced, because Al2O3 interlayers only slightly modify electrical properties of the InP/Sm2O3-based capacitors. The presentation of results must be improved.

1) The title should not contain abbreviatures such as MOS and ALD.

2) All abbreviatures should be explained, when they appear in the text. See, ALD in the abstract, for example.

3) The introduction is quite narrow and does not describe the soundness of topic. It is mainly focused on some technical characteristics. What authors can bring to a broad audience about the development of nanomaterials for MOS capacitors?

4) I'm dissapointed to read "indium phosphorous oxide" rather than indium phosphate.

5) In the experimental section the XPS standard samples and their origin should be presented.

6) The interpretation of XPS results raises questions: equations 1 and 2, why metallic indium and aluminum are present in your samples? Is it possible that the appearance of aluminum and indium phosphates is due to the reaction of InP and Al2O3 in the presense of oxygen and traces of H2O?

7) In Figure 4, what is the level of probability that was used to fit the linear region?

8) In the section 3.3 Electrical Properties of InP-MOS capacitors, the obtained electrical characteristics should be compared with literature.

Round 2

Reviewer 1 Report

In my initial review of the manuscript 3 major concerns were pointed out. I regret to report that the authors failed to address 2 of them in a satisfactory manner, therefore I have to suggest that the manuscropt be rejected.

More specifically, the manuscript has not received the extensive editing needed in order to be easily understood by the readership, in my opinion. The English remains poor and it makes it very hard to follow the reasoning of the authors in many parts of the paper.

More importantly, my initial concerns about the originality of the presented work and its comparison to the existing literature have not been adequately addressed. In their response to the comments, the authors have created a table that compares certain aspects of their work with 2 other published results. This comparison is not enough, in my opinion. Firstly, they do not compare all relevant quantities, mainly the desity of interface states, which they claim is the highlight of their work. As I pointed out in my initial review, there is published work (using different dielectrics) that show a smaller number of Dit.

Moreover, the table and the references quoted in their response to the review has not been inserted anywhere in the manuscript, as far as I can tell! Nor is there any text elaborating on the originality of the presented research.

Reviewer 2 Report

The changes are not highlighted in the revised text. Nor the cover letter contains summary of changes or numbers of pages and lines where the changes have been made.

Round 3

Reviewer 1 Report

The authors have answered all of my comments in a satisfactory manner with their latest version of the manuscript.

Moreover, the english in the text has been substantially improved.

I believe that the manuscript is now ready for publication.